# Effects of Quercitrin on PRV-Induced Secretion of Reactive Oxygen Species and Prediction of lncRNA Regulatory Targets in 3D4/2 Cells

**DOI:** 10.3390/antiox11040631

**Published:** 2022-03-25

**Authors:** Qiuhua Wang, Xiaodong Xie, Qi Chen, Shouli Yi, Jiaji Chen, Qi Xiao, Meiling Yu, Yingyi Wei, Tingjun Hu

**Affiliations:** College of Animal Science and Technology, Guangxi University, Nanning 530005, China; qiuhuawang@gxu.edu.cn (Q.W.); 1718304007@st.gxu.edu.cn (X.X.); chenqi@st.gxu.edu.cn (Q.C.); yishouli@st.gxu.edu.cn (S.Y.); 2118302002@st.gxu.edu.cn (J.C.); 1918393054@st.gxu.edu.cn (Q.X.); yumeiling@gxu.edu.cn (M.Y.); weiyingyi@gxu.edu.cn (Y.W.)

**Keywords:** quercitrin, pseudorabies virus, lncRNA, oxidative stress, ROS

## Abstract

Quercitrin is a kind of flavonoid that is found in many plants; it has good antioxidant activity, and can regulate oxidative stress induced by Pseudorabies virus (PRV)-infected cells. In this study, the secretion of reactive oxygen species (ROS) induced by PRV infection was detected by flow cytometry, and RNA expression profiles of the 3D4/2 cells were produced and analyzed by sequenced GO (Gene Ontology) and KEGG (Kyoto Encyclopedia of Genes and Genomes); the sequencing results were verified by RT-qCR. The results showed that the secretion of ROS induced by PRV infection in 3D4/2 cells could be significantly decreased by quercitrin. The differentially expressed 1055 mRNA, 867 lncRNA, 99 miRNA, and 69 circRNA were detected between the control group and the PRV infection group. The differentially expressed 1202 mRNA, 785 lncRNA, 115 miRNA, and 79 circRNA were found between the PRV+ quercitrin group and the control group. The differentially expressed 357 mRNA, 69 lncRNA, 111 miRNA, and 81 circRNA were obtained between the PRV+ quercitrin group and the PRV group. The significantly differentially expressed mRNAs were mainly involved in cell metabolism, regulatory protein phosphorylation, protein phosphorylation, antioxidation, regulatory phosphorylation, and so on. Among them, the mRNAs related to antioxidant response and oxidative stress were thioredoxin-interacting protein (TXNIP) and nitric oxide synthase 2 (NOS2). According to the network diagram of lncRNA–miRNA–mRNA, two targeted miRNA (ssc-miR-450c-3p and novel-m0400-3p) relationships with TXNIP and NOS2 were screened. This study provides a scientific foundation for further research for the function of quercitrin in anti-virus-induced oxidative stress.

## 1. Introduction

Pseudorabies virus (PRV) was first isolated in 1902 and has spread all over the world since the 1980s [1]. It is a highly contagious virus with a high fatality rate among pigs, and can cause great economic losses every year around the world. Apart from pigs, it can survive in a variety of animals, such as mice, sheep, goats, cats and rabbits [2]. It has also been reported to be a potential zoonotic pathogen [3]. In addition, studies have shown that viruses such as highly pathogenic avian influenza virus [4], pseudorabies virus [5,6] and other viruses can induce oxidative stress in cells, resulting in the imbalance of intracellular oxidative systems and antioxidant systems, as well as the production of intracellular reactive oxygen species (ROS) and their oxidative intermediates, resulting in the destruction of lipids, proteins, nucleic acids, and other substances, further leading to apoptosis [7,8], inflammatory reaction [4,5], immunosuppression [9], and so on.

Studies have reported that some plant extracts have antiviral, anti-inflammatory and antioxidant effects, such as *andrographolide* [10], Raw Rehmannia *Radix Polysaccharide* [11], total flavonoids of *Spatholobussuberectus* Dunn [12], and flavonoids from *Polygonum hydropiper* [13]. Quercitrin (3,3,4,5,7-pentahydroxyflavone) is a polyphenolic flavonoid compound found in a variety of plants, such as *Polygonum hydropiper*, and some evidence shows that quercitrin possesses anti-inflammatory, antioxidant, antiviral, and antiproliferative properties, and has beneficial roles in ROS scavenging [14,15]. Quercitrin has been used to promote growth and to reduce heat stress in chickens [16], and helps heterophils to perform more effective and efficient oxidative killing of microbes due to its effect on ROS generation [17].

Long non-coding RNA (lncRNA) plays an important regulatory role in the transcription of viral and host genes, and in host antiviral responses [18,19]. Jin [20] reported that the silencing of LncA02830 could up-regulate the transcription levels of IRF3 and IFNβ, as well as MX1, and inhibit the replication of PRV-II. This can be a potential target for antiviral therapy. In addition, some plant extracts can regulate the level of oxidative stress by regulating the expression of lncRNA in cells. For example, chlorogenic acid increased retinal ganglion cell viability and decreased ROS level and RGC apoptosis after oxidative stress injury by lncRNA TUG1/Nrf2 pathways [21]. However, there are few reports on how plant extracts regulate virus infection by regulating lncRNA. In this paper, we investigate the changes in PRV-induced secretion of reactive oxygen species, and the intracellular RNA expression profile in 3D4/2 cells, and predict the relationships of lncRNA, mRNA, and miRNA, which will provide a theoretical basis for clinical practice by clarifying the antiviral mechanism of quercitrin.

## 2. Materials and Methods

### 2.1. Reagent, Cells and Viruses

The quercitrin (Purity 98%) was purchased from Solarbio (Beijing Soleibao Technology Co., Ltd., Beijing, China).

PK-15 cells (porcine kidney 15), 3D4/2 cells (a porcine lung cell line) and PRV were provided by the Laboratory of Animal Infectious Disease at the College of Animal Science and Technology at Guangxi University. The cells were cultured in Dulbecco’s Modified Eagle high glucose medium (DMEM; Gibco, Grand Island, NY, USA) supplemented with 10% fetal bovine serum (FBS; Gibco, USA), penicillin (100 IU/mL), and streptomycin (100 mg/mL) at 37 °C in an atmosphere of 5% CO_2_.

### 2.2. Assessment of Cell Viability

The cell viability analysis of quercitrin was detected using the CCK-8 assay (Beyotime Biotechnology, Shanghai, China). This experiment included the control group, a DMSO group and quercitrin groups (12.5, 25, 50, 100, 200, 400, 800 and 1600 μM). The quercitrin group was marked as Q group. The 3D4/2 cells with a density of 1 × 10^5^ cells/mL were seeded in 96-well plates and cultured for 12 h. Then, the supernatant was removed, the cells were treated with 0.1%DMSO and different concentrations of quercitrin, the cells were sub-cultured for 6 h, 12 h, 24 h, 36 h or 48 h, and then the culture medium was replaced with 100 μL 10% CCK-8 and incubated in 5% CO_2_ at 37 °C for 1 h. The optical density was measured at the 450 nm wavelength with a multifunctional microplate reader (Tecan Infinite M200 Pro, Grodig, Austria).

### 2.3. Titrating TCID50 of PRV

The PK15 cells were seeded in a 96-well plate (1 × 10^4^ cells·/well) and cultured in 5% CO_2_ at 37 °C with 5%-FBS-DMEM for 12 h. The culture medium was discarded and the cells were washed three times by 0.1 mol/L phosphate buffered solution (PBS, Beijing Soleibao Technology Co., Ltd., Beijing, China). Except for the cell control groups, the cells were adsorbed with PRV at a dilution of 10^−1^ to 10^−10^ for 1 h, and the plates were shaken once every 15 min. Then, the inoculum was removed, the cells were washed three times with 0.1 mol/L PBS, and the cells were further cultured in 5% CO_2_ at 37 °C with 2%-FBS-DMEM. The cytopathic effect (CPE) was observed and recorded every 12 h, until no new CPE appeared.

### 2.4. Detection of Intracellular ROS by Flow Cytometry

The experiment was conducted using six groups including negative control groups, as shown in Table 1. The symbols “+” and “−” represent adding and not adding, respectively, ROS probe, PRV and quercitrin. The ROS detection kit (Shanghai Beyotime Biotechnology Co., Ltd., Shanghai, China) was used to detect changes in intracellular ROS. First, 2.5 mL 3D4/2 cells of 1 × 10^5^ cells·mL^−1^ diluted with 5% DMEM were added to the 6-well plate and were cultured in the 37 °C incubator of 5%CO_2_ for 12 h; then, the cells were washed three times with 0.1 mol/L of PBS. The cells were inoculated with PRV, except the cell control and negative control groups. After 1 h, the supernatant was discarded and cells washed with 0.1 mol/L PBS three times. Then, cells were treated with quercitrin and further cultured 24 h. DCFH-DA was added and the cells were collected according to the instructions of the ROS kit; then, the cell suspension (50 μL) was subjected to flow cytometry (Attune Nxt, Thermo, Singapore) analysis.

### 2.5. Sequencing and Analysis RNA

On the basis of the above experiments, the proper quercitrin concentration of 25 μM was chosen in the following studies. The experiments were divided into three groups: the control group (C), the PRV group (P), and the PRV+ Quercitrin group (P + Q), with three replicates in each group. The 3D4/2 cells of 2.5 mL at 1 × 10^5^ cells·mL^−1^ were added to the 6-well plate. After culturing in a 37 °C incubator of 5% CO_2_ for 12 h, the culture medium was discarded and cells were washed with PBS three times. The P group and P + Q group were inoculated with 0.01 MOI (multiplicity of infection) of PRV and incubated at 37 °C in 5% CO_2_ for 1 h; then, the cells were treated and cultured for another 24 h. After 24 h of co-culture, the cells were collected and the total RNA was extracted according to the instructions of TRizol (TAKARA BIO INC, Dalian, China.

The total RNA was sent to Guangzhou Chideo Biotechnology Co., Ltd. (Guangzhou, China) to construct an RNA library for mRNA, lncRNA, miRNA and circRNA library sequencing (Illumina HiSeqTM 4000,Gene Denovo Biotechnology Co., Ltd., Guangzhou, China). According to the sequencing results, the target genes of mRNA and lncRNA that were differentially expressed were analyzed by Gene Ontology (GO) enrichment analysis with GO-seq software, and the differential mRNA with the same expression trend in the C group, P group, or P + Q group was clustered by STEM software. RNAplex software was used to predict and construct the network diagram of the interaction between mRNA and lncRNA.

### 2.6. Verification of Differential Expression RNA

In the trend analysis of differential expression (DE) RNA, the RNA was selected for fluorescence quantitative PCR verification (RT-qPCR), and the relative expression was calculated by the 2^−ΔΔΔct^ method. The sequence of RNA primers is shown in Table 2.

### 2.7. Calculation and Statistical Analysis

The results are presented as the means ± SD (standard deviation). All the data were statistically analyzed by SPSS 22.0 (SPSS, Chicago, IL, USA). One-way ANOVA was used to test the main effect. When there were significant differences (*p* < 0.05), and there were extremely significant differences (*p* < 0.01), the group means were further compared using Duncan’s multiple range test, and the results are presented in a histogram prepared in GraphPad Prism 6.0.

## 3. Results

### 3.1. Effect of Quercitrin on the 3D4/2 Cells’ Viability

As Figure 1 shows, there was no difference in cell viability between the DMSO group and the control group; the cell viability of 1600 μM and 800 μM groups was significantly lower than that of the control group at 6 h, 12 h, 24 h, and 36 h (*p* < 0.01); the cell viability of the 400 μM group was extremely significantly lower than that of the control group at 12 h and 24 h (*p* < 0.01); treatment with quercitrin at the concentrations of 25 μM, 50 μM, and 100μM significantly increased cell viability at 12 h, 24 h, and 36 h (*p* < 0.01). In addition, after treatment with quercitrin at 12 h and 48 h, the cell viability of the 25 μM group was significantly higher than that of the control group (*p* < 0.05).

### 3.2. The TCID_50_ of PRV

The cytopathic changes of each group were observed every 12 h, and no new CPE was observed at 56 h. According to the results of Reed–Muench calculations, the TCID_50_ of PRV was 10^−5.643^/0.1 mL (Table 3).

### 3.3. Effect of PRV Infection on ROS Production in 3D4/2 Cells

The percentages of fluorescent cells in the P group and the P + Q group were significantly higher than that in the control group (*p* < 0.05) (Figure 2), while the percentages of fluorescent cells in the quercitrin 25, 50 and 100 μM groups were significantly lower than that in the P group (*p* < 0.05), but there were no differences among the quercitrin treatment groups.

### 3.4. The Differential Expression of mRNA, lncRNA, miRNA and circRNA

About 983,000 to 757,000 raw reads and 981,000 to 755,000 clear reads per sample were obtained and the error rate was about 0.28% to 0.19% (Appendix A). Deseq was used to analyze the differential expression of RNA; a fold change (FC) greater than 1.5 and *p* < 0.05 were used as the screening conditions for differential expression with mRNA, lncRNA, miRNA and circRNA. PRV infection significantly up-regulated 746 mRNAs, 581 lncRNAs, 66 miRNAs and 28 circRNAs, while it down-regulated 309 mRNAs, 286 lncRNAs, 33 miRNAs and 41 circRNAs compared to the control group. Compared with the control group, 692 mRNAs, 476 lncRNAs, 76 miRNAs, and 41 circRNAs were up-regulated, while 510 mRNAs, 309 lncRNAs, 39 miRNAs, and 38 circRNAs were down-regulated in the P + Q group. Moreover, in the P + Q group, compared with the P group, 147 mRNAs, 26 lncRNAs, 64 miRNAs and 43 circRNAs were up-regulated and 210 mRNAs, 43 LncRNAs, 47 miRNAs and 28 circRNAs were down-regulated (Figure 3A,D,G,J); the most differentially expressed top 20 genes in each group were mapped as heat maps (Figure 3C,F,I,L). In addition, through the statistics of differentially expressed mRNA, lncRNA, miRNA, and circRNA by a Wayne diagram (Figure 3B,E,H,K), it was found that some of the differentially expressed mRNA, lncRNA and miRNA were differentially expressed in the three groups. According to the statistics of the 14 mRNAs differentially expressed among the three groups, it was found that there were two genes related to oxidative stress and anti-inflammatory stress, namely thioredoxin-interacting protein (TXNIP) and nitric oxide synthase 2 (NOS2) (Appendix A).

### 3.5. GO Analyses of DE mRNAs

GO annotation mainly includes three parts: biological process (BP), cellular component (CC), and molecular function (MF). In order to further understand the biological function of differentially expressed mRNAs, the mRNAs of three groups were analyzed by GO (Figure 4). Between the C group and the P group, the top 20 terms of mRNA differentially expressed in GO annotation include the regulation of cell metabolism, regulation of protein phosphorylation, regulation of protein phosphorylation, regulatory phosphorylation, and so on. Between the C group and the P + Q group, the top 20 terms differentially expressed mRNA annotated by GO include the regulation of cell metabolism, regulated protein phosphorylation, and so on. Between the P group and the P + Q group, the top 20 terms of differentially expressed mRNA annotated by GO include protein targeting, protein transport, protein transport into the nucleus, and oxygen homeostasis.

### 3.6. KEGG Analyses of DE mRNAs

The KEGG shows that the differentially expressed mRNA enrichment pathways between the C group and the P group, and between the C group and the P + Q group, are the MAPK signal pathway, the TNF signal pathway, and the TGF-β signal pathway. The differentially expressed mRNA enrichment pathways between the P group and the P + Q group are RNA transport, the AMPK signal pathway, and the PPAR signal pathway (Figure 5).

### 3.7. Trend Analysis of mRNAs

Trend analysis and STEM analysis were conducted on 1839 DE mRNAs of three experience groups (C, P, P + Q), and eight trend cluster maps were obtained. Then, 290 genes were down-regulated (Figure 6A), 132 genes were up-regulated (Figure 6H), 140 genes were down-regulated and then up-regulated (Figure 6C), 174 genes were up-regulated and then down-regulated (Figure 6F), 228 genes were down-regulated and then stabilized (Figure 6B), 126 genes were stabilized and then down-regulated (Figure 6D), 63 genes were stabilized and then up-regulated (Figure 6E), and 686 genes were up-regulated and then stabilized (Figure 6G).

### 3.8. Analysis of DE mRNAs Related to Oxidative Stress and Antioxidant Stress

According to the mRNA sequencing results of NOS2, TXNIP, PSEN1, HIF1A, PSIP1, SETX, cytochrome c oxidase subunit 8A (COX8A), COX2, SOD1, MST1R, and other genes related to oxidative stress and antioxidation in the control group, P group and P + Q group (Figure 7A), it was found that, compared with the control group, the mRNA expression of COX8A and SOD1 in the P group increased, while the mRNA expression of PSIP1, SETX, and TXNIP decreased, and the differential expression of other genes was not significant (Figure 7B). In the P + Q group, the mRNA expression of SOD1 increased, the mRNA expression of TXNIP decreased, and there was no difference in the expression of other genes (Figure 7C). When compared with the PRV group, the mRNA expression of NOS2, TXNIP, COX8A, and MST1R decreased significantly and the mRNA expression of HIF1A, SETX, PSEN1, PSIP1, and COX2 increased significantly in the P + Q group (Figure 7D).

### 3.9. Validation of lncRNA and mRNA

Six RNAs were randomly selected from trend cluster maps profile2 (Figure 6C) and profile5 (Figure 6F) for RT-qPCR verification. The results showed that the relative expression of lncRNA or mRNA of RT-qPCR was consistent with the sequencing results, as shown in Figure 8.

### 3.10. The Interaction between lncRNAs, mRNAs and miRNAs in PRV + Quercitrin Group and PRV Group

Some of the mRNAs were selected in trend clustering (Figure 6C,F) to find the differential lncRNA with cis- and trans-antisense of these mRNAs, and then find the miRNA that had a targeted relationship with mRNA and LncRNA. Cytoscape software was used to construct an lncRNA–miRNA–mRNA network diagram (Figure 9). In the network diagram, mRNA, lncRNA, and miRNA are represented by circles, triangles and rectangles, with red for up-regulation and blue for down-regulation. Black solid lines are used to represent the antisense relationship, black dashed lines are used to represent the trans relationship, blue solid lines are used to represent the cis, and red solid lines are used to represent the targeting relationship between the miRNA and mRNA. There are three known lncRNAs, identified in the network diagram as ENSSSCT00000081255, ENSSSCT00000070508, and ENSSSCT00000066182, and six new lncRNAs, with names beginning with MSTRG. The mRNAs related to oxidation and metabolic function have cytochrome c oxidase subunit 8A(COX8A) and cytochrome P450, family 51, subfamily A(CYP51A1). The mRNA of thioredoxin-interacting protein (TXNIP) and nitric oxide synthase 2 (NOS2) are related to the antioxidant response and oxidative stress. Among them, there are eight miRNAs that targeted NOS2, which are novel-m0696-5p, novel-m0514-5p, ssc-miR-450c-3, novel-m0609-5p, miR-940-z, miR-127-x, miR-19-zand novel-m0400-3p, respectively, and the lncRNA with a trans relationship with NOS2 was MSTRG.6502.1. There are seven miRNAs that targeted TXNIP; they are miR-106-z, novel-m0400-3p, novel-m0152-3p, miR-93-z, miR-1304-y, novel-m0514-5p, ssc-miR-450c-3p, and the lncRNA with a trans relationship with MSTRG.287.6. The results show that miRNA ssc-miR-450c-3p and novel-m0400-3p had a targeting relationship with NOS2 and TXNIP.

## 4. Discussion

The cell viability experiment might suggest that the viability of 3D4/2 was related to the concentration and the treatment time of quercitrin. High-concentration quercitrin (≥400 µM) can inhibit cell proliferation; the inhibition peaked 24 h after treatment, and after 36 h the inhibition weakened and the cell viability increased. Quercitrin at a low concentration (≤200 µM) can promote cell proliferation, and the concentration of 25~100 µM was suitable for the cell. This phenomenon indicates that the content of quercitrin may be decreased with the increase of treatment time. Therefore, at 48 h, the concentration of quercitrin in the 200 µM and 400 µM groups was reduced to a concentration that could promote cell proliferation, but the quercitrin concentration in the 800 µM group was reduced to a concentration that could not inhibit cell proliferation.

Some results show that PRV caused apoptosis in a porcine kidney cell line, PK15, and induced expressions of proapoptotic Bcl family proteins, specific DNA damage sensors and the phosphorylation of histone H2AX, and subsequently activated the expressions of checkpoint kinase 1/2 and proapoptotic p53; this confirmed that oxidative stress and free radicals arising from PRV infection cause DNA damage, consequently triggering apoptosis [6]. Our results show that the secretion of reactive oxygen species in 3D4/2 cells infected by PRV increased, while the reactive oxygen species decreased after treatment with different doses of quercitrin. These findings suggest that quercitrin may have a potential antiviral effect and reduce the production of reactive oxygen species induced by PRV. In order to further study the mechanism of quercitrin regulating the decrease in reactive oxygen species secretion in 3D4/2 cells, RNA-seq sequencing was carried out.

The results of GO and KEGG enrichment of RNA-seq showed that there were significant differences in mRNA and lncRNA expression between the quercitrin virus group and the virus control group, and the differentially expressed signal pathways were mainly concentrated in cell metabolism, regulation of protein phosphorylation, protein phosphorylation, regulation of phosphorylation, and other signal pathways, including the differential expression of genes related to antioxidation and other signal pathways.

Among them, thioredoxin-interacting protein (TXNIP), as an important binding partner for thioredoxin (TRX) that can mediate oxidative stress, inhibited cell proliferation, and induced apoptosis by inhibiting the function of the thioredoxin system and has been confirmed to be an important regulator for redox-related signal transduction [22]. It is an important regulator of glucose and lipid metabolism through pleiotropic actions, including the regulation of β-cell function, peripheral glucose uptake, adipogenesis, and substrate utilization, and can be one of the therapeutic targets for diabetes [23], with wide range of functions in cardiovascular diseases, neurodegenerative diseases, cancer, and other diseases [24]. Cho observed that TXNIP expression was rapidly down-regulated in RPE cells under oxidative stress and that RPE cell proliferation was decreased [25].

Pathogens and inflammatory cytokines activate NOS2 and generate high concentrations of NO through the activation of inducible nuclear factors, including NF-κB, resulting in large numbers of free radicals [26,27,28]. In addition, NOS2 may be one of the biomarkers and predictors of poor prognosis and an ideal target for cancer therapy [29].

There were seven miRNAs and one lncRNA targeted to TXNIP, eight miRNAs and one lncRNA targeted to NOS2, and miRNAs ssc-mir-450c-3p and novel-m0400-3p were targeted to NOS2 and TXNIP. The results indicate that miRNA ssc-mir-450c-3p and novel-m0400-3p may be potential targets of quercitrin in regulating reactive oxygen species secretion induced by PRV infection.

## 5. Conclusions

Quercitrin can decrease the level of ROS produced by 3D4/2 cells infected by PRV and alleviate the oxidative stress caused by PRV infection. This study provided differential RNA expression profiles of PRV-infected immune cells and quercitrin-treated PRV-infected immune cells and predicted the relationship between lncRNA and mRNA in PRV-infected immune cells treated with quercitrin; two potential targets miRNA ssc-miR-450c-3p and novel-m0400-3p were screened out, which provides a basis for the further study of antiviral function of quercitrin.

## Figures and Tables

**Figure 1 antioxidants-11-00631-f001:**
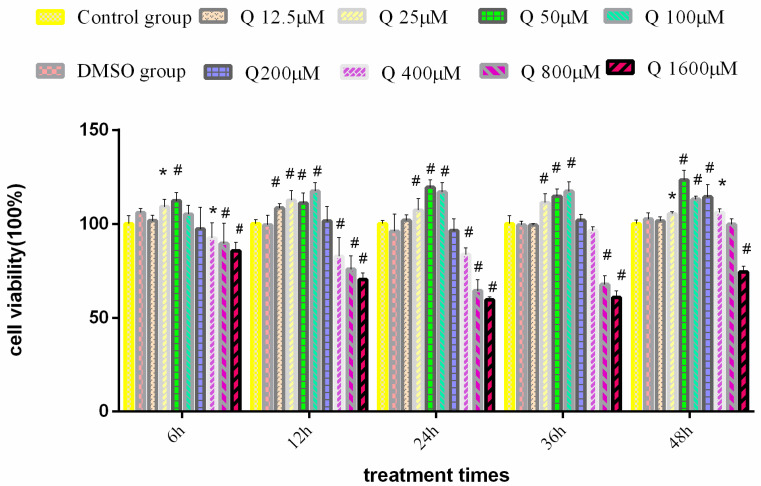
Effect of quercitrin on the 3D4/2 cells’ viability (*n* = 8): Bars with * indicate significant difference compared with control group (*p* < 0.05). Bars with ^#^ indicate extremely significant difference compared with control group (*p* < 0.01).

**Figure 2 antioxidants-11-00631-f002:**
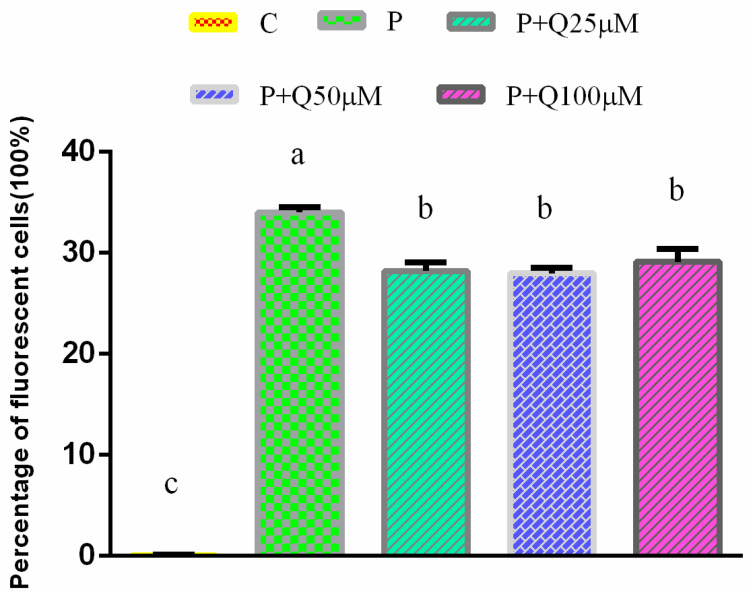
Percentage of fluorescent cells detected by flow cytometry (mean ± S.D, *n* = 3). Bars with different letters (a, b, c) are statistically different (The same letter means no significant difference, and different letters mean significant difference *p* < 0.05).

**Figure 3 antioxidants-11-00631-f003:**
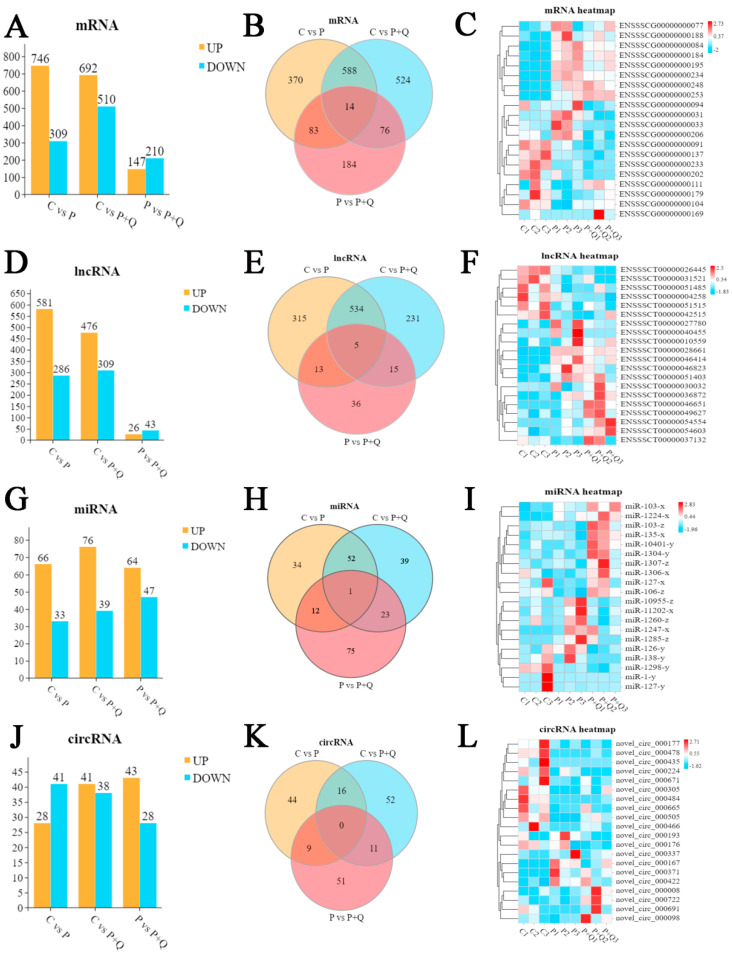
Differential expression RNA analysis. Histogram, Wayne chart and heat map generated by hierarchical clustering analysis of differentially expressed mRNA (**A**–**C**), LncRNA (**D**–**F**), miRNAs (**G**–**I**), and circRNAs (**J**–**L**). (FC ≥ 1.5, *p* < 0.05.).

**Figure 4 antioxidants-11-00631-f004:**
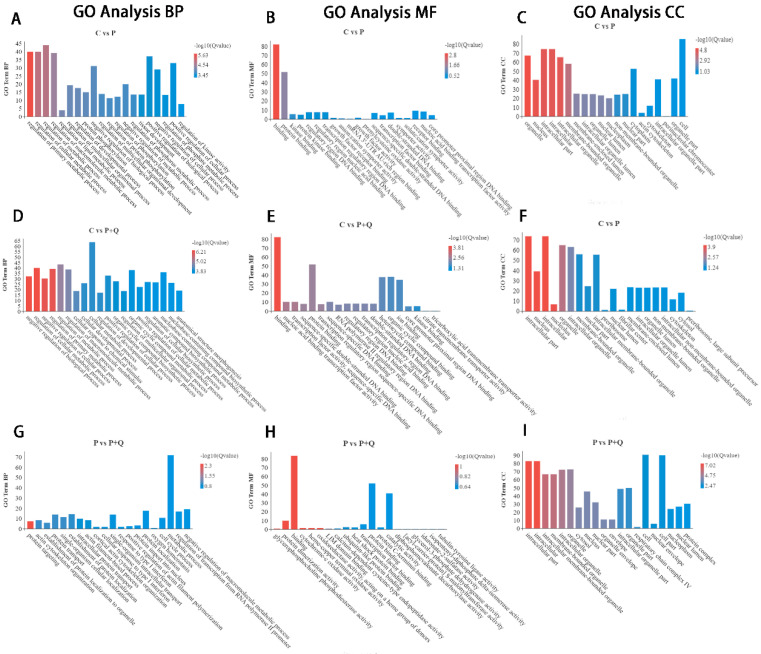
Gene Ontology term analysis of DE mRNAs. C group and P group: top 20 enrichment genes in BP (**A**), MF (**B**), CC (**C**). C group and P + Q group: top 20 enrichment genes in BP (**D**), MF (**E**), CC (**F**). P group and P + Q group: top 20 enrichment genes in BP (**G**), MF (**H**), CC (**I**).

**Figure 5 antioxidants-11-00631-f005:**
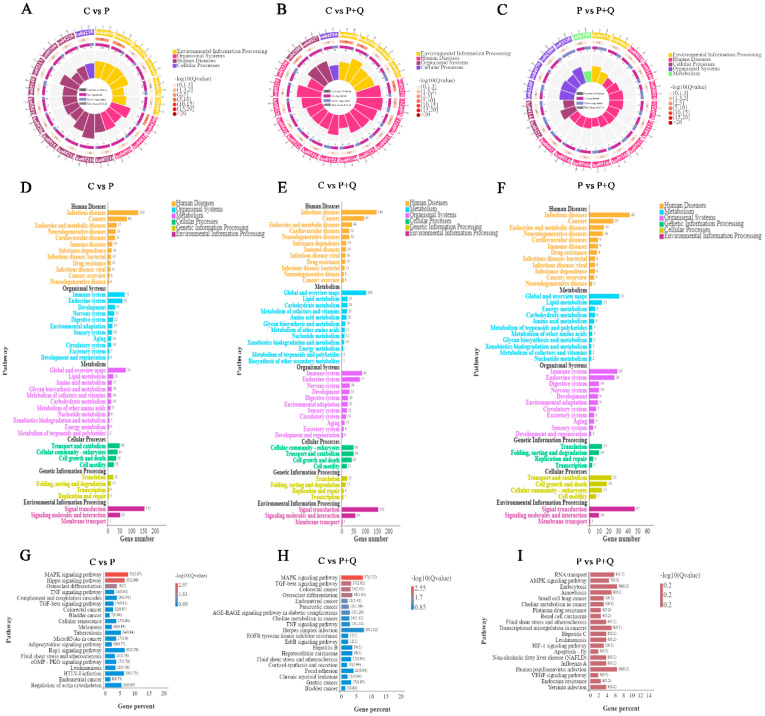
KEGG analyses of DE mRNAs. Enrichment and screening of KEGG pathway in the C group and the P group resulted in 316 signaling pathways on top of KEGG enrichment MAPK, Hippo, osteoclast differentiation, TNF signaling pathway, and other signaling pathways (**A**,**D**,**G**). A total of 321 signal pathways were obtained by enrichment and screening of the KEGG pathway in the C group and the P + Q group, which, on top of KEGG enrichment, mainly involved MAPK, TGF-βα and others signaling pathways (**B**,**E**,**H**). Moreover, 249 signal pathways were obtained by enrichment and screening of the KEGG pathway in the P group and the P + Q group, on top of KEGG enrichment, which mainly involved RNA transport, AMPK, HIF-1, the VEGF signaling pathway and other signal pathways (**C**,**F**,**I**).

**Figure 6 antioxidants-11-00631-f006:**
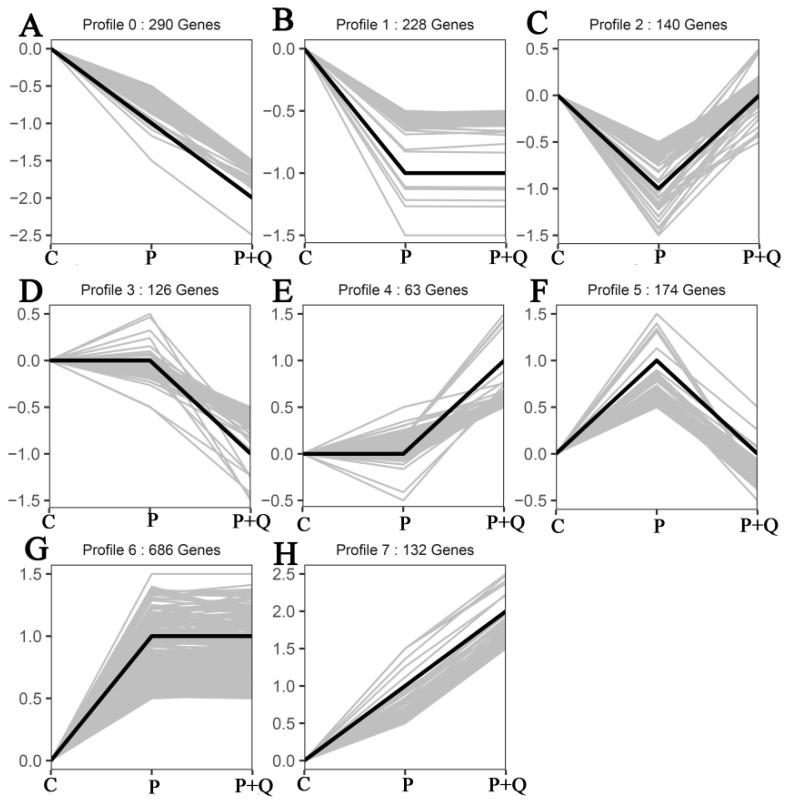
Trend cluster maps from DE mRNA by STEM analysis. Between the three groups, DE mRNA were down-regulated between (**A**), up-regulated (**H**), down-regulated and then up-regulated (**C**), up-regulated and then down-regulated (**F**), down-regulated and then stabilized (**B**), stabilized and then down-regulated (**D**), stabilized and then up-regulated (**E**), up-regulated and then stabilized (**G**).

**Figure 7 antioxidants-11-00631-f007:**
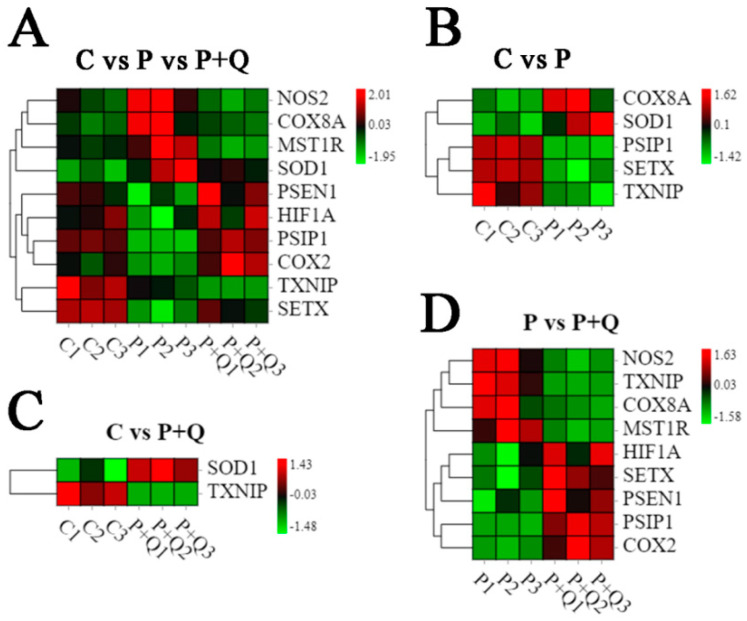
Analysis of differential expression of genes related to oxidative stress and antioxidant stress. Heat map representing the level of expression, Comparison of expression levels of 10 genes in three groups (**A**), Comparison of expression levels of 5 genes in C and P groups (**B**), Comparison of expression levels of 2 genes in C and P + Q groups (**C**), Comparison of expression levels of 9 genes in P and P + Q groups (**D**).Red represents higher expression, and green represents lower expression.

**Figure 8 antioxidants-11-00631-f008:**
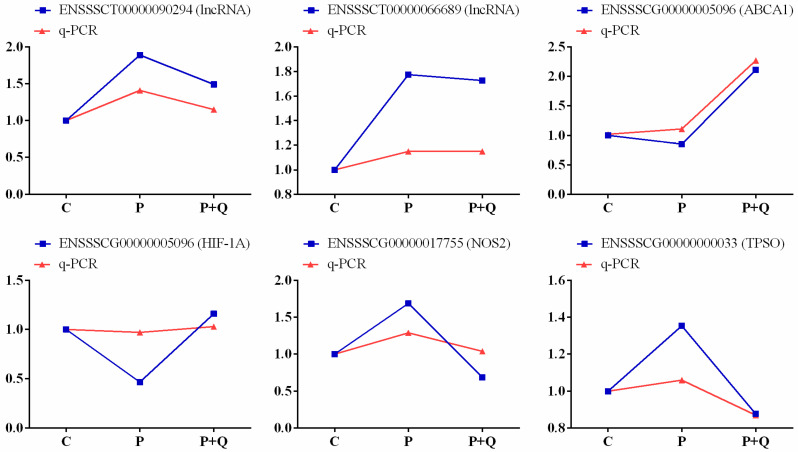
Sequencing FPKM relative value and q-PCR relative expression curve.

**Figure 9 antioxidants-11-00631-f009:**
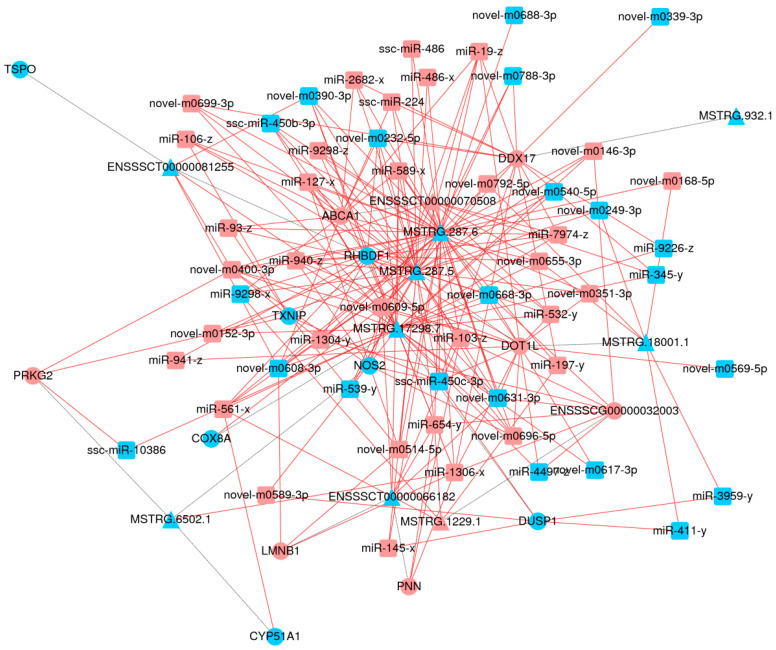
LncRNA, mRNA and miRNA interaction network diagram (P vs. P + Q).

**Table 1 antioxidants-11-00631-t001:** Experimental grouping of ROS detection.

Group	ROS Probe	PRV	Quercitrin
Negative control	−	−	−
Control(C)	+	−	−
P	+	+	−
P + Q 25 μM	+	+	+
P + Q 50 μM	+	+	+
P + Q 100 μM	+	+	+

**Table 2 antioxidants-11-00631-t002:** Primer information of RNA.

Target Gene	Accession	Sequence (5′–3′)	Amplicon Size (bp)	Genbank
lncRNA1 F		TGAATGCGGTTAGTCCTTGGTTCC	148	ENSSSCT00000090294
lncRNA1 F		GACGACGACACAGGCAGAGTAAG		
lncRNA2 R		GAGATTCGTGGCTGCTGTGAGTAG	108	ENSSSCT00000066689
lncRNA2 R		GAAGCGTGGGCGAGGAAGAAC		
ABCA1 F	100152112	CGCCTCCTTCGTGTTCAAGATCC	82	ENSSSCG00000005423
ABCA1 R		ACTGCCATTGATGCCGATGAAGAG		
HIF1A F	396696	CATTTCCATCTCCTCCCCACGTA	170	ENSSSCG00000005096
HIF1A R		ACTCAAAGCGACAGATAACACA		
NOS2 F	396859	TACCCCACCAGACGAGCTTC	122	ENSSSCG00000017755
NOS2 R		CTATCTCCTTTGTTACCGCTTCC		
TPSO F	396592	CTCACGCAATGTCCTCGGAA	134	ENSSSCG00000000033
TPSO R		TCATGTAGGAGCCATACCCCAT		
β-actin F	414396	GATGAGATTGGCATGGCTTT	101	ENSSSCG00000007585
β-actin R		CACCTTCACCGTTCCAGTTT		

**Table 3 antioxidants-11-00631-t003:** Recorded results of half-lethal dose determination (*n* = 8).

PRV Dilution	Number of CPE Holes Present	Number of No CPE Holes Appear	Accumulative Total		Percentage of CPE Holes Present (%)
			No CPE	CPE	
10^−1^	8	0	0	42	100
10^−2^	8	0	0	34	100
10^−3^	8	0	0	26	100
10^−4^	8	0	0	18	100
10^−5^	8	0	0	10	100
10^−6^	1	7	7	2	22.2
10^−7^	1	7	14	1	6.7
10^−8^	0	8	22	0	0
10^−9^	0	8	30	0	0
10^−10^	0	8	38	0	0

## Data Availability

Some or all data, models, or code generated or used during the study are available from the corresponding author by request Gene data can be downloaded at https://ngdc.cncb.ac.cn/gsa, and the gene accession number is CRA006088 and CRA006080.

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
