# Peer review of "Effects of Quercitrin on PRV-Induced Secretion of Reactive Oxygen Species and Prediction of lncRNA Regulatory Targets in 3D4/2 Cells"

_antioxidants, 2022, doi:10.3390/antiox11040631_

Round 1
Reviewer 1 Report
antioxidants-1614568
Title: „Effects of quercetin on PRV-induced secretion of reactive oxygen species and prediction of lncRNA regulatory targets in 3D4/2 cells”
Recommendation: Minor revision
In the paper „Effects of quercetin on PRV-induced secretion of reactive oxygen species and prediction of lncRNA regulatory targets in 3D4/2 cells” by Qiuhua Wang et al. the authors demonstration the that the secretion of ROS induced by PRV infection in 3D4/2 cells could be significantly decreased by quercetin.
-> The experiments are planned and designed correctly.
-> The Materials and Methods, Results and Discusion of this work are presented correctly.
-> Does the Author’s checked the cell cultures against mycoplasma?
-> The Author’s should enlarge the font on the x and y axis (Figure 5-6).
Author Response
- Does the Author’s checked the cell cultures against mycoplasma?
Response: In the experiment, the cell cultures were regularly checked against mycoplasma by PCR, and before sending samples for sequencing , the cell cultures were also checked against mycoplasma. The PCR primer sequence are P1 “ 5' –GGGCCAAGAGGTTGTA- 3'” and P2 “5' -CCTTCGCCTATTGGTG-3'”, the positive control amplified fragment is about 400bp.
- The Author’s should enlarge the font on the x and y axis (Figure 5-6).
Response: x and y axis of the fig.4-5 (original order was Figure 5-6) were enlarged and the information is readable when the manuscript was enlarged.

Reviewer 2 Report
The work described in the present manuscript is consistent with the scope of the journal.
Authors described the effect of an antioxidant agent, quercetin, and inspected the relationships of lncRNA and antioxidant stress targets in pseudorabies virus infected 3D4/2 cells a porcine lung cell line.
However, the paper needs many improvements to be suitable for publication, therefore major revisions are suggested. Manuscript must be carefully revised since there are numerous typographical errors. English must be revised also.
Some points should be address prior to a possible publication, specifically:
Major comments:
- Along all manuscript, correct “quercitrin” to “quercetin”.
- Rewrite the first sentence of the abstract. It doesn’t make sense…
- Section 2.4: The description of experimental groups is quite exhaustive along the manuscript, repeated numerous times mainly in figures’ legends. I suggest adding a table in the Materials and Methods section where the authors define the designation of each group and the specific conditions, assigned for example with + or – symbols, if present or in absence, for example. Then, they can just mention this table when needed, especially in figures legends.
- Section 2.5: Designation of experimental groups described in this section is different from that defined in section 2.4, particularly P+Q and PRV+Que, which I believe are the same. Please uniformize. Like this is quite confusing…
- Line 3.1, please define in the title the type of “activity” inspected. The same in the title of Table 2 (line 157)
- Regarding the data presented in Table 2, authors must discuss why the activity is only significantly different from the control after 48h, for concentrations of 200 and 400 um (considering that lower concentration showed different results for earlier timepoints). Also, it must be discussed why for concentrations between 200-1600 um, the values are lower than the control, in opposition to that found for lower concentrations.
- Legend of Figure 1, remove details of concentrations and composition of the experimental groups (“notes”). This information must be referred preferentially in the MM section. The same must be applied to the other figures of the manuscript.
- Table 2 and Figure 1 summarize the same information. Please chose one to be in the main text and the other can be moved to ESI. Regarding the statistic, all significances are just p<0.05 or lower? This must be clarified since I believe that there are different levels of significance.
- Figure 3 must me move to ESI, I believe the most relevant information is summarized in Fig. 2.. Tables 4 and 5 can also be moved.
- Figure 5, please increase the size, it is not readable (horizontal layout)
- Figure 6, specify data described in in section (A-I). The same in fig. 7 an 8.
Minor comments:
- Line 12, remove the space after “Genomes”
- Line 17, remove repeated comma
- Line 20, add a space after final mark.
- Line 55, replace “play” by “plays”
- Line 83, please specify in the Materials and Methods section, the tested concentrations
- Line 90, add a space after 37ºC.
- Line 116, move 25 uM to line 114, after “concentration”
- Line 120, “2” in “CO2” must be subscripted
- Line 120, add a space after MOI.
- Line 134, add a space after “expression”
- Line 135, add a space after “verification”
- Line 147, replace “Result” by “Results”
- Table 3, please define the meaning of “CPE”; replace “not” by “no”
- Line 183, remove “which”
Author Response
1.Along all manuscript, correct “quercitrin” to “quercetin”.
Response: All “quercetin” are corrected in the manuscript, and were marked in red.
- Rewrite the first sentence of the abstract. It doesn’t make sense…
Response: The first sentence of the abstract was rewriten.
3.Section 2.4: The description of experimental groups is quite exhaustive along the manuscript, repeated numerous times mainly in figures’ legends. I suggest adding a table in the Materials and Methods section where the authors define the designation of each group and the specific conditions, assigned for example with + or – symbols, if present or in absence, for example. Then, they can just mention this table when needed, especially in figures legends.
Response: The description of experimental groups was revised, which presented in table form. (table 1).
4.Section 2.5: Designation of experimental groups described in this section is different from that defined in section 2.4, particularly P+Q and PRV+Que, which I believe are the same. Please uniformize. Like this is quite confusing…
Response: In revised manuscript, PRV and PRV+ Quercitrin groups were uniformized the abbreviation of P and P+Q.
5.Line 3.1, please define in the title the type of “activity” inspected. The same in the title of Table 2 (line 157)
Response: The “activity” was replaced by “viability”, it means cell proliferative capacity.
6.Regarding the data presented in Table 2, authors must discuss why the activity is only significantly different from the control after 48h, for concentrations of 200 and 400 um (considering that lower concentration showed different results for earlier time points). Also, it must be discussed why for concentrations between 200-1600 um, the values are lower than the control, in opposition to that found for lower concentrations.
Response: The discussion cell viability assessment was added in the part of “discussion”.
7.Legend of Figure 1, remove details of concentrations and composition of the experimental groups (“notes”). This information must be referred preferentially in the MM section. The same must be applied to the other figures of the manuscript.
Response: The “notes” under the figure were removed and the details were described in line 81, 82, 83. Other “notes” under the figures were replaced by specific information.
8.Table 2 and Figure 1 summarize the same information. Please chose one to be in the main text and the other can be moved to ESI. Regarding the statistic, all significances are just p<0.05 or lower? This must be clarified since I believe that there are different levels of significance.
Response: Table 2 was deleted, the figure 1 was retained and added statistics of extremely significant p<0.01.
9.Figure 3 must me move to ESI, I believe the most relevant information is summarized in Fig. 2.. Tables 4 and 5 can also be moved.
Response: Figure 3, tables 4 and 5 were deleted.
10.Figure 5, please increase the size, it is not readable (horizontal layout)
Response: The x and y axis were enlarged and the information is readable when the manuscript was enlarged.
11.Figure 6, specify data described in in section (A-I). The same in fig. 7 an 8.
Response: Specify data were adding under the figure (3, 4, 5, 7 ) which original order was fig. 4, fig. 5. fig. 6. fig. 8. The fig.6 which original order was fig.7 data was described in line 241~247.
Minor comments:
1.Line 12, remove the space after “Genomes”
2.Line 17, remove repeated comma
3.Line 20, add a space after final mark.
4.Line 55, replace “play” by “plays”
5.Line 83, please specify in the Materials and Methods section, the tested concentrations
6.Line 90, add a space after 37ºC.
7.Line 116, move 25 uM to line 114, after “concentration”
8.Line 120, “2” in “CO2” must be subscripted
9.Line 120, add a space after MOI.
10.Line 134, add a space after “expression”
11.Line 135, add a space after “verification”
12.Line 147, replace “Result” by “Results”
13.Table 3, please define the meaning of “CPE”; replace “not” by “no”
14.Line 183, remove “which”
Response: As required, the tested concentrations in line 83 were specified in line 81-83 which marked red, the “CPE” was defined in line 98 and marked red, and other “Minor comments” had revised as required which marked red.

Round 2
Reviewer 2 Report
The authors have addressed my concerns in the present revision. The modifications performed in the manuscript clarified most of the non-well explained topics and significantly improved the quality of the manuscript. Just a last comment, I believe that the first sentence of the abstract was not rewritten. After addressing the proposed modification, I consider the paper suitable for publication.
Author Response
Comment: 1. I believe that the first sentence of the abstract was not rewritten. After addressing the proposed modification, I consider the paper suitable for publication.
Response: The first sentence of abstract was changed into “Quercitrin is a kind of flavonoids that are broadly found in many plants, which has good antioxidant activity, and it can regulate oxidative stress induced by Pseudorabies virus (PRV) infected cells.”, meanwhile the parts marked in red in first revision are uniformly changed to black.
